# $A^2R^2$: ADVANCING IMG2LATEX CONVERSION VIA VISUAL REASONING WITH ATTENTION-GUIDED REFINEMENT

## ABSTRACT

Img2LaTeX is a practically important task that involves translating mathematical expressions and structured visual content from images into LaTeX code. In recent years, vision-language models (VLMs) have achieved remarkable progress across a range of visual understanding tasks, largely due to their strong generalization capabilities. However, despite initial efforts to apply VLMs to the Img2LaTeX task, their performance remains suboptimal. Empirical evidence shows that VLMs can be challenged by fine-grained visual elements, such as subscripts and superscripts in mathematical expressions, which results in inaccurate LaTeX generation. To address this challenge, we propose $A^2R^2$: **A**dvancing Img2LaTeX Conversion via Visual **R**easoning with **A**ttention-Guided **R**efinement, a framework that effectively integrates attention localization and iterative refinement within a visual reasoning framework, enabling VLMs to perform self-correction and progressively improve LaTeX generation quality. For effective evaluation, we introduce a new dataset, *Img2LaTex-Hard-1K*, consisting of 1,100 carefully curated and challenging examples designed to rigorously evaluate the capabilities of VLMs within this task domain. Extensive experimental results demonstrate that: (1) $A^2R^2$ significantly improves model performance across various evaluation metrics spanning both textual and visual levels; (2) Increasing the number of inference rounds yields notable performance gains, underscoring the potential of $A^2R^2$ in test-time scaling scenarios; (3) Ablation studies and further evaluations confirm the effectiveness of our approach and the synergy of its core components during inference.

## 1 INTRODUCTION

In modern applications, users frequently interact with chat agents and consume research content where mathematical expressions and structured information must be represented in LaTeX format. This demand highlights the need for models that can accurately convert screenshots or images into their corresponding LaTeX source code. Existing approaches primarily rely on convolutional neural networks (CNNs) or Vision Transformer (ViT)-based architectures, which are fine-tuned on large-scale datasets specifically curated for this task (Jiang et al., 2025; Wang et al., 2019b; Dosovitskiy et al., 2021; Wang & Liu, 2021; Wang et al., 2019a). However, these models typically rely heavily on large-scale training data and lack the capacity for human-like reasoning and self-correction when faced with mismatches or prediction errors.

Recently, vision-language models (VLMs) have demonstrated strong potential in multimodal understanding, particularly in tasks that require reasoning over image-text interactions (Zhang et al., 2024b; Du et al., 2022; Ghosh et al., 2024; Caffagni et al., 2024; Zhang et al., 2024a; Yin et al., 2024). With the increasing availability of such models, VLMs are emerging as promising candidates for tackling the Img2LaTeX task. Nonetheless, prior studies evaluating their performance on this task reveal notable limitations, indicating that there remains substantial room for improvement (Roberts et al., 2024). One possible explanation is the reliance on direct evaluation of VLMs, which may overlook the potential advantages of leveraging their visual reasoning capabilities during inference.

To address the aforementioned limitations, we construct a more challenging subset from the Im2LaTeX-100K dataset (Deng et al., 2017), selecting approximately 1,100 difficult examples using

Figure 1: An illustration of the $A^2R^2$ framework applied to the Img2LaTeX task. Unlike direct inference, which yields an incorrect result, $A^2R^2$ incorporates multiple reasoning steps into the inference process. By leveraging iterative refinement, the framework progressively enhances the output, ultimately generating the correct LaTeX expression.

a combination of metric-based filtering and evaluations conducted by multi-modal large language models (MLLMs). This new dataset, *Img2LaTex-Hard-1K*, is specifically designed to stress-test the capabilities of current VLMs under more demanding conditions.

Inspired by how visual reasoning emulates human-like thinking through self-correction and iterative refinement (Tan et al., 2025; Bi et al., 2025; Chen et al., 2024a; Zhang et al., 2025; OpenAI, 2025; Google, 2024; Xu et al., 2025; Wang et al., 2025), we propose a novel training-free plug-in framework, $A^2R^2$: **A**dvancing Img2LaTeX Conversion via Visual **R**easoning with **A**ttention-Guided **R**efinement. Our proposed framework enhances VLM performance on the Img2LaTeX task by integrating attention-based localization with iterative refinement guided by visual feedback. As illustrated in Figure 1, $A^2R^2$ consists of four core stages: (1) Generation: the VLM generates an initial LaTeX hypothesis from the input image; (2) Rendering and Comparison: the predicted LaTeX is rendered into an image and visually compared against the input to identify discrepancies, which are then used to elicit feedback; (3) Attention Localization and Feedback Verification: attention mechanisms guide the model to focus on the mismatched regions, while the system assesses the reliability of the feedback; (4) Refinement: the LaTeX output is updated based on the verified feedback, and the process iterates from stage (2), allowing the model to perform self-correction through visual reasoning.

In summary, our key contributions are as follows:

(1) We propose $A^2R^2$, a novel visual reasoning framework that integrates attention-based localization and iterative self-refinement to enhance VLM performance on the Img2LaTeX task, all within a training-free paradigm.

(2) Extensive experiments demonstrate that $A^2R^2$ consistently outperforms other baselines. Moreover, increasing the number of inference steps yields notable improvements, supporting the effectiveness of test-time scaling.

(3) Ablation studies and human evaluations provide further evidence of the practical benefits of the proposed framework, revealing strong synergy among its core components during inference.

(4) We introduce *Img2LaTex-Hard-1K*, a dataset of 1,100 challenging samples curated to rigorously benchmark modern VLMs, which exhibit substantially greater capabilities than prior generations.

## 2 RELATED WORKS

### 2.1 IMG2LATEX

Img2LaTeX is a well-established task involving the conversion of an image containing LaTeX-rendered content into its corresponding textual LaTeX source. This task is crucial in academic and educational contexts, where accurate transcription of mathematical and scientific notation is essential (Peng et al., 2021; Kayal et al., 2023; Wang & Shan, 2020). Prior work primarily adopts computer vision-based architectures for LaTeX recognition (Jiang et al., 2025; Wang et al., 2019a;

Wang & Liu, 2021). While these models are effective, they typically rely on large-scale annotated datasets and often struggle with visually complex inputs. More recently, vision-language models have been applied to this task, but findings indicate that their performance remains limited in this domain-specific setting (Roberts et al., 2024). Motivated by these challenges, we propose a novel approach that incorporates visual reasoning to enhance VLM performance on the Img2LaTeX task.

## 2.2 VISUAL REASONING

With the emergence of the test-time scaling paradigm, researchers increasingly adopt training strategies such as supervised fine-tuning (SFT) and group relative policy optimization (GRPO) to enhance the reasoning capabilities of large language models (LLMs) (Shao et al., 2024b; Yeo et al., 2025; Muennighoff et al., 2025). These methods support extended chain-of-thought reasoning and self-correction during inference, showing promise in domains like mathematical problem solving and code generation (DeepSeek-AI et al., 2025; Mei et al., 2025; OpenAI et al., 2024a; OpenAI, 2025; Google, 2024). Similar efforts in vision-language models (VLMs) aim to enable long-form multimodal reasoning. Recent work leverages image-text training to support extended reasoning chains (Shen et al., 2025; Dong et al., 2025; Xu et al., 2025; Thawakar et al., 2025; Wang et al., 2025), while others incorporate object localization to ground attention in evidence-rich image regions (Gao et al., 2025; Shao et al., 2024a). Additional approaches explore multi-agent self-correction (Li et al., 2025a). These advancements motivate our integration of visual reasoning to improve VLM performance on the Img2LaTeX task.

## 3 IMG2LATEX-HARD-1K

The Im2LaTeX-100k dataset introduced by Deng et al. (2017) remains a foundational benchmark for LaTeX recognition. However, our preliminary analysis shows that state-of-the-art vision-language models (VLMs) exceed $90\%$ accuracy on roughly $75\%$ of instances, suggesting that much of the dataset lacks sufficient difficulty for meaningful evaluation. This saturation limits the ability to assess model capabilities and differentiate performance in more challenging scenarios.

To address this limitation and enable more discriminative evaluation of current VLMs, we introduce *Img2LaTex-Hard-1K*, a curated subset designed to stress-test contemporary models. The curation combines quantitative performance-based filtering with qualitative assessments of visual complexity, targeting instances that reveal weaknesses in mathematical reasoning and fine-grained visual understanding. *Img2LaTex-Hard-1K* serves two main goals: offering a more rigorous benchmark for model comparison and facilitating the analysis of failure modes to guide future research.

We construct the *Img2LaTex-Hard-1K* benchmark by evaluating diverse open-weight VLMs across multiple scales and architectures, combining textual similarity metrics (m-ROUGE, BLEU-4, Edit Distance) with visual fidelity scores from GPT-4O-MINI, and aggregating them into weighted instance-level difficulty scores to guide final data selection. The detailed construction pipeline is fully presented in Appendix C.

## 4 METHODOLOGY

Traditional vision-language models (VLMs) often struggle to capture fine-grained visual details in LaTeX expressions, resulting in subtle yet critical errors during LaTeX generation. To address this limitation, we propose the $A^2R^2$ framework, which introduces an iterative visual reasoning process. By rendering predictions and comparing them against input images, the model autonomously detects and corrects errors through attention-guided localization and targeted refinement. The $A^2R^2$ framework operates in five stages: (1) Generation, (2) Rendering, (3) Comparison, (4) Attention Localization and Feedback Verification, and (5) Refinement. Each stage is described in detail below.

### 4.1 GENERATION

In the initial stage, a vision-language model (VLM) is employed to produce an initial LaTeX prediction. Given an input image $I$ and a generation prompt $P_{\text{generation}}$, the model analyzes the visual

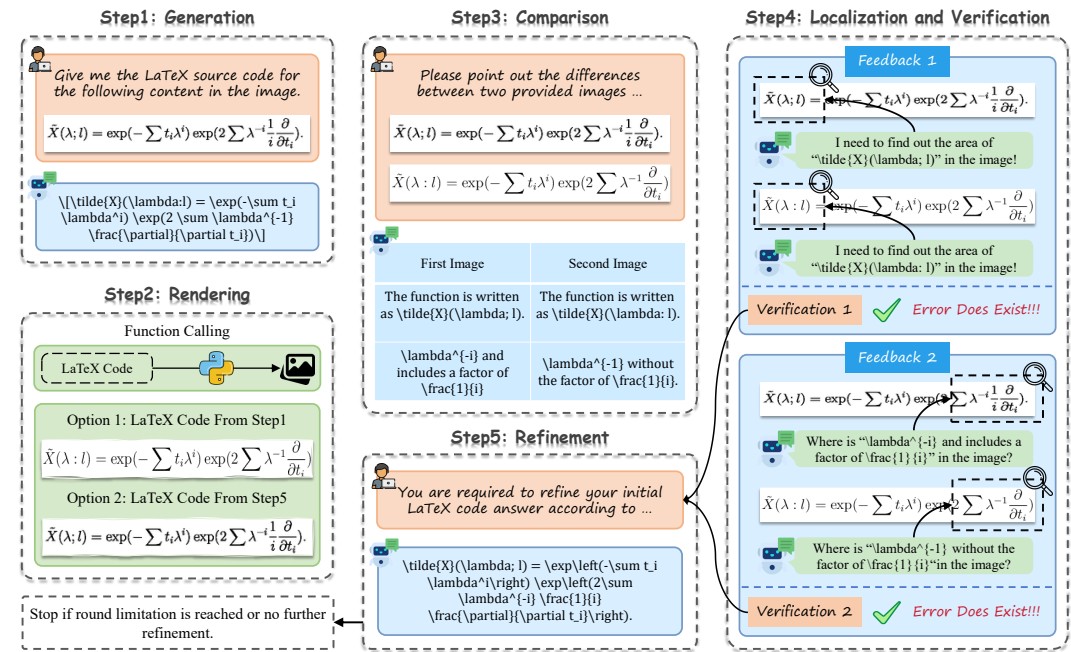

Figure 2: A detailed illustration of how the $A^2R^2$ framework solves the Img2LaTeX task. The process consists of multiple stages: generation, rendering, comparison, attention localization, feedback verification, and refinement. These stages form a recurrent structure that enables iterative improvement. This extended reasoning mechanism supports test-time scaling and effectively corrects initial errors, ultimately producing the correct output.

content and generates the corresponding LaTeX sequence. This process is formalized as:

$$L = \text{VLM}(I, P_{\text{generation}}),$$

where $L$ denotes the LaTeX output generated by the model.

### 4.2 RENDERING

Once the initial LaTeX output is generated, it is rendered into an image $I'$ using external tools such as *pdflatex* in conjunction with *ImageMagick*. For the first round of inference, the input LaTeX code corresponds to the output from the *Generation* stage. In subsequent rounds, the input is taken from the output of the *Refinement* stage.

### 4.3 COMPARISON

In the *Comparison* stage, the rendered image $I'$ is paired with the original input image $I$, and both are fed back into the model. The vision-language model now acts as a visual difference evaluator, identifying discrepancies between the original and generated images. This process is formalized as:

$$D = \text{VLM}(I, I', P_{\text{comparison}}),$$

where $P_{\text{comparison}}$ denotes a prompt specifically designed to guide the model in identifying and describing differences between $I$ and $I'$ in a structured format. The resulting output $D$ represents the model-generated feedback, which is subsequently used for verification and refinement.

### 4.4 ATTENTION LOCALIZATION AND FEEDBACK VERIFICATION

Although vision–language models possess the ability to identify differences between two images, they remain prone to hallucinations, particularly when their performance on Img2LaTeX conversion is limited. Therefore, after detecting discrepancies between the original image $I$ and its rendered counterpart $I'$, we employ an attention-based localization mechanism to highlight regions with high

attention. These regions are assumed to capture potential semantic or structural mismatches. We then extract two focused subregions from both $I$ and $I'$ to enable more fine-grained verification.

Let the textual prompt fed to the model be a sequence $\boldsymbol{\tau} = (\tau_1, \tau_2, \ldots, \tau_n)$ of $n$ tokens that describe the mathematical content to be verified. For each token $\tau_i$, we extract attention weights from a specified range of cross-attention layers, spanning from $l_{\text{start}}$ to $l_{\text{end}}$, inclusive. Each layer comprises $H_{\text{head}}$ attention heads. The attention map corresponding to token $\tau_i$ at layer $l$ and head $h$ is represented as $W_i^{(l,h)} \in \mathbb{R}^{H \times W}$.

We first average across all heads and all selected layers to compute a unified attention map for each token $\tau_i$:

$$\tilde{W}_i = \frac{1}{(l_{\text{end}} - l_{\text{start}} + 1) \cdot H_{\text{head}}} \sum_{l=l_{\text{start}}}^{l_{\text{end}}} \sum_{h=1}^{H_{\text{head}}} W_i^{(l,h)}. \tag{1}$$

Subsequently, we average over all $n$ tokens to obtain the final attention matrix:

$$A = \frac{1}{n} \sum_{i=1}^{n} \tilde{W}_i, \; A_{u,v} = \frac{1}{n} \sum_{i=1}^{n} \tilde{w}_{i,(u,v)}, \; u = 1, \ldots, H, \; v = 1, \ldots, W. \tag{2}$$

This yields $A \in \mathbb{R}^{H \times W}$, where $H$ and $W$ denote the spatial dimensions (in patch units) of the image feature map. Each entry in $A$ represents the average attention across the selected layers, heads, and tokens, thereby highlighting how the model aligns the textual prompt with different image regions.

Since the values in the attention matrix typically do not reach 1, we normalize them into an 8-bit grayscale image to prepare the matrix for contour detection:

$$A_{\text{norm}} = 255 \cdot \frac{A - \min(A)}{\max(A) - \min(A)}. \tag{3}$$

We threshold the normalized attention matrix at the 75$^{\text{th}}$ percentile to isolate top-attention regions:

$$B(i,j) = \begin{cases} 255 & \text{if } A_{\text{norm}}(i,j) \geq \tau, \\ 0 & \text{otherwise} \end{cases} \quad \text{where } \tau = \text{Percentile}(A_{\text{norm}}, 75), \; \forall(i,j). \tag{4}$$

This produces a binary map $B \in \{0, 255\}^{H \times W}$, where pixels with high attention are white and others are black. We then extract the contours $\mathcal{C} = \{\mathcal{C}_1, \ldots, \mathcal{C}_k\}$ from $B$ using standard external contour detection and select the largest contour based on area:

$$\mathcal{C} = \text{Contours}(B), \quad \mathcal{C}^* = \arg\max_{\mathcal{C}_i \in \mathcal{C}} \text{Area}(\mathcal{C}_i). \tag{5}$$

To obtain the final region, we first dilate the largest contour $\mathcal{C}^*$ with a rectangular structuring element $K$ of size $3 \times 3$:

$$\mathcal{C}_{\text{dil}} = \mathcal{C}^* \oplus K. \tag{6}$$

We then compute the bounding box of $\mathcal{C}_{\text{dil}}$ to extract the corresponding subregion $R$:

$$(x, y, w, h) = \text{Bounding}(\mathcal{C}_{\text{dil}}), \quad R = \{(i,j) \in I \mid x \leq j < x + w, \; y \leq i < y + h\}. \tag{7}$$

Through this process, we obtain two regions cropped from the original input image and the rendered image, denoted as $R$ and $R'$. These regions are then fed into the model for self-verification:

$$D' = \text{VLM}(D, R, R', P_{\text{verification}}).$$

This attention-guided localization and verification step enables the model to focus on high-attention regions, enhancing its robustness in filtering out hallucinated or incorrect feedback.

## 4.5 Refinement

In the final step, we utilize the cropped regions from the original image $R$ and the rendered image $R'$, together with the previously identified correct difference description $D'$, to guide the model in revising the current LaTeX generation $L$.

A refinement prompt $P_{\text{refinement}}$ is designed to ensure that the model modifies only the erroneous part of $L$ while preserving its correct components:

$$L^{\text{updated}} = \text{VLM}(L, R, R', D', P_{\text{refinement}}).$$

| Base Model | Inference Method | Textual Metrics | | | | | Visual Metrics | |
|---|---|---|---|---|---|---|---|---|
| | | ROUGE-1↑ | ROUGE-2↑ | ROUGE-L↑ | BLEU-4↑ | Edit Distance↓ | Match↑ | CW-SSIM↑ |
| LLAMA-3.2-11B-VISION-INSTRUCT | Direct Prompting | 84.14 | 68.59 | 83.69 | 64.83 | 27.45 | 89.66 | 87.00 |
| | Chain-of-Thought Prompting | 78.11 | 61.30 | 77.30 | 50.75 | 41.28 | 89.52 | 86.97 |
| | Best-of-N (N = 2) | 84.51 | 68.74 | 83.92 | 64.98 | 27.10 | 89.92 | 87.18 |
| | Best-of-N (N = 4) | 84.76 | 68.87 | 84.04 | 65.13 | 26.84 | 90.17 | 87.26 |
| | Best-of-N (N = 8) | 84.98 | 69.01 | 84.11 | 65.23 | 26.63 | 90.24 | 87.38 |
| | $A^2R^2$ (Ours) | **90.87** | **73.13** | **89.21** | **70.41** | **20.12** | **93.75** | **93.46** |
| QWEN2.5-VL-32B-INSTRUCT | Direct Prompting | 79.47 | 60.59 | 77.94 | 55.21 | 31.35 | 90.86 | 89.00 |
| | Chain-of-Thought Prompting | 75.62 | 57.59 | 74.45 | 51.32 | 46.08 | 89.80 | 87.61 |
| | Best-of-N (N = 2) | 79.72 | 60.81 | 78.12 | 55.36 | 30.95 | 91.07 | 89.23 |
| | Best-of-N (N = 4) | 79.92 | 61.05 | 78.28 | 55.50 | 30.37 | 91.24 | 89.39 |
| | Best-of-N (N = 8) | 80.04 | 61.19 | 78.39 | 55.61 | 30.02 | 91.33 | 89.52 |
| | $A^2R^2$ (Ours) | **86.92** | **66.17** | **83.45** | **62.32** | **22.87** | **94.16** | **94.58** |

Table 1: Performance of two vision-language models on the filtered *Img2LaTex-Hard-1K* dataset, evaluated across seven metrics spanning both textual and visual dimensions. The best score for each model under each metric is highlighted in bold red.

Here, $L^{\text{updated}}$ denotes the updated LaTeX code after correcting the identified error. Following this refinement, the process returns to step (2) to verify whether additional discrepancies remain. If so, the model repeats steps (2) to (5) iteratively:

$$L^{(t+1)} = \text{REFINE}(L^{(t)}, I, I').$$

This self-refinement loop continues until no new differences are detected or a predefined iteration limit $T_{\max}$ is reached. The final output is given by:

$$L^* = L^{(T)}, \quad \text{where } T = T_{\max} \text{ or } \text{diff}(I, I'^{(T)}) = \emptyset.$$

## 5 EXPERIMENTS SETUP

### 5.1 DATASET

We use our curated *Img2LaTex-Hard-1K* dataset, which comprises 1,100 images containing LaTeX content.

### 5.2 VISION-LANGUAGE MODELS

Our proposed method relies on identifying salient regions across image pairs by accessing attention weights during inference. To facilitate this, we adopt open-weight vision-language models that expose internal attention mechanisms. For our main experiments, we select two models with distinct architectural designs and parameter scales:

- QWEN2.5-VL-32B-INSTRUCT (Bai et al., 2025): A 32-billion-parameter model from the QWEN2.5 family, representing a large-scale transformer-based architecture.
- LLAMA-3.2-11B-VISION-INSTRUCT (Meta AI, 2024): An 11-billion-parameter model from the LLAMA3.2 family, chosen to evaluate the effect of reduced model capacity.

### 5.3 METRICS

We evaluate the similarity between the generated LaTeX source code and the ground-truth label from both textual and visual perspectives.

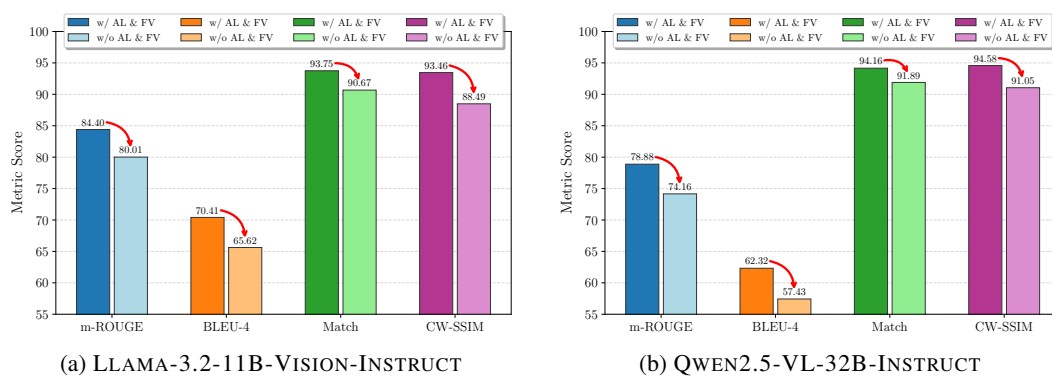

(a) LLAMA-3.2-11B-VISION-INSTRUCT  (b) QWEN2.5-VL-32B-INSTRUCT

Figure 3: Ablation results for two vision-language models across four metrics demonstrate the impact of **A**ttention **L**ocalization and **F**eedback **V**erification (AL & FV) on overall performance. Darker bars indicate the full model, while lighter bars represent variants with AL and FV removed. *m-ROUGE* denotes the mean of *ROUGE-1*, *ROUGE-2*, and *ROUGE-L* scores.

**Textual Metrics.**    To assess the fidelity of the generated LaTeX sequences at the token and character levels, we employ the following standard text generation metrics:

- *ROUGE* (Lin, 2004): A recall-oriented metric that measures n-gram overlap between the generated sequence and the reference.
- *BLEU-4* (Papineni et al., 2002): A precision-oriented metric that evaluates the overlap of up to 4-grams.
- *Edit Distance* (Ristad & Yianilos, 1998): Computes the minimum number of edits needed to match the ground-truth sequence.

**Visual Metrics.**    To evaluate the visual fidelity of the rendered LaTeX outputs, we compare the predicted and ground-truth images using the following image-level metrics:

- *Match*: Measures the proportion of identical pixels between the predicted and ground-truth renderings.
- *CW-SSIM* (Sampat et al., 2009): Evaluates structural similarity in the complex wavelet domain, robust to minor distortions.

## 5.4 IMPLEMENTATION DETAILS

We obtain all model weights from the official repositories hosted on HuggingFace. All experiments are conducted on a machine equipped with four NVIDIA A100 80GB GPUs. To ensure consistency and reproducibility, we use the default inference settings provided by each model. Further details and an introduction to the baseline are provided in Appendix D and Appendix E, respectively.

## 6 EXPERIMENT RESULTS

For the main experiments, we evaluate 160 test instances using two models, comparing our method against three baselines. To ensure fairness, we cap our method at two inference rounds, keeping the average token count comparable to Best-of-N ($N = 8$). As shown in Table 1, $A^2R^2$ consistently outperforms other baselines across both models and all textual and visual metrics.

Under the LLAMA-3.2-11B-VISION-INSTRUCT model, CoT prompting introduces interpretability but consistently degrades performance across all metrics, suggesting that verbose reasoning may hinder generation quality in multimodal settings. Best-of-N sampling yields slight gains as $N$ increases from 2 to 8, but improvements remain modest, indicating limited textual diversity despite increased computational cost.

In contrast, our framework delivers substantial improvements. *ROUGE* increases by approximately 6 points, *Edit Distance* decreases to 20.12, and visual metrics improve significantly, with *Match*

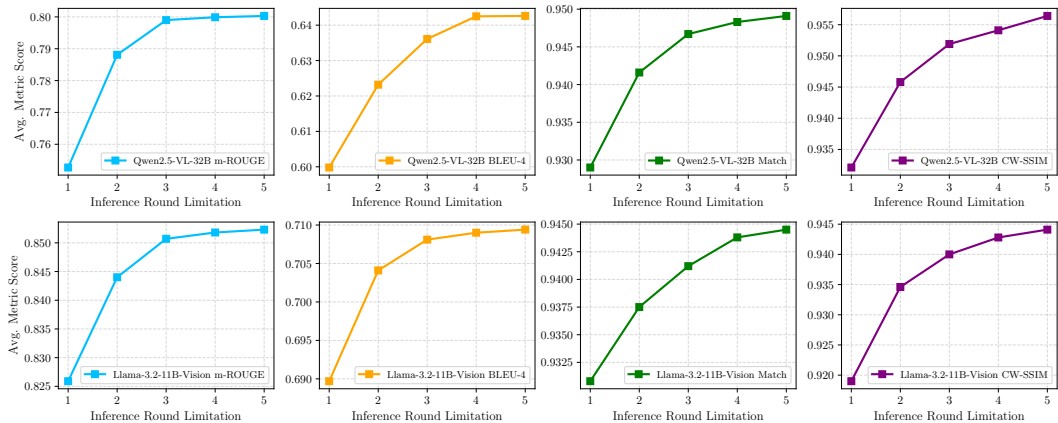

Figure 4: Performance of two vision-language models under four evaluation metrics using the proposed method with an expanded inference limitation round. Results demonstrate that extending test-time execution leads to improved performance across both models.

rising from 89.66 to 93.75 and *CW-SSIM* from 87.00 to 93.46. These results highlight $A^2R^2$'s ability to maintain structural fidelity and cross-modal coherence. Similar patterns are observed with the QWEN2.5-VL-32B-INSTRUCT model. CoT again leads to performance degradation, while our method achieves consistent gains across all metrics. *BLEU-4* improves from 55.21 to 62.32, and visual alignment strengthens further, with *Match* reaching 94.16 and *CW-SSIM* 94.58, the highest across all settings.

Overall, these results validate the effectiveness and robustness of our proposed framework. In contrast to simple prompt engineering or sampling heuristics, $A^2R^2$ fundamentally enhances the model's ability to interpret, align, and verbalize visual information. The improvements span both text-level and image-level evaluations across different models, demonstrating the generalizability of our framework.

## 7 ABLATION STUDY

Our framework introduces attention-based visual reasoning, which localizes crucial regions and integrates feedback verification during refinement. This design mitigates hallucinated differences incorrectly identified by the model, thereby improving both accuracy and reliability. This raises the central question: *How much do attention localization and feedback verification help mitigate the negative effects of visual hallucination?* To answer this, we conduct ablation experiments with both models, comparing refinement with and without these components.

| Base Model | Round 1 | Round 2 | Round 3 |
|---|---|---|---|
| LLAMA-3.2-11B-VISION-INSTRUCT | 21.21% | 25.83% | 27.56% |
| QWEN2.5-VL-32B-INSTRUCT | 17.78% | 22.04% | 23.77% |

Table 2: Hallucination rates during the comparison step across Rounds 1, 2, and 3 using our proposed method, highlighting the importance of attention localization and feedback verification.

As shown in Figure 3, relying only on textual feedback without attention localization and cropped verification leads to a clear performance drop across four metrics. Both models show declines of over 4 points in *m-ROUGE* and *BLEU-4*, while *Match* and *CW-SSIM* fall by 2.5–3.5 points. These results indicate that direct refinement based solely on textual feedback provides limited benefit. The degradation stems from the base models' limited comparison abilities, where unverified feedback often introduces errors by altering correct content into incorrect predictions.

Table 2 further supports this finding by reporting hallucination rates during the first three refinement rounds, measured using the more reliable GPT-4O model (OpenAI et al., 2024c). The results confirm the ablation analysis and highlight the necessity of structured verification to guard against errors introduced by hallucinated feedback.

## 8   TEST-TIME SCALING

Our framework employs iterative refinement, where feedback is generated over multiple rounds by identifying differences between two input images. This process embodies the idea of test-time scaling: allowing more refinement rounds increases inference time but enhances the model's ability to detect discrepancies and improve outputs.

We evaluate two vision-language models under different round limits using four metrics, with results shown in Figure 4. Performance consistently improves as the number of rounds increases, though gains diminish beyond three rounds. For instance, the improvement from three to five rounds is notably smaller than that from one to two, suggesting that models gradually reach their reasoning capacity and fewer errors remain for correction.

Overall, the upward trends validate the test-time scaling property of our method and highlight the effectiveness of iterative visual reasoning. These results also suggest that stronger base models could provide more accurate feedback, enabling further improvements through refinement.

## 9   HUMAN EVALUATION

Evaluating Img2LaTeX is inherently complex, as both textual and visual fidelity must be considered. The metrics in Table 1 ensure fair comparisons by using the same base models, but they are sensitive to stylistic variations in LaTeX expressions. For example, syntactic differences such as using "*text*" may alter token-based scores without affecting the rendered image.

To better capture human preferences, we conduct a human evaluation with three computer science graduate students proficient in LaTeX. We randomly sample 100 outputs from two models under different inference methods. Each annotator assigns a score from 0 to 5 (with 0.5 increments) based on visual similarity to the ground truth, with 5 indicating indistinguishable outputs aside from padding or minor formatting differences. Scores are averaged across annotators for each method, and results are shown in Figure 5.

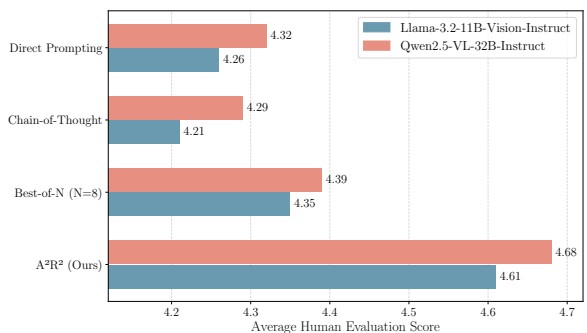

Figure 5: Human evaluation scores of different inference methods with two tested vision-language models, our method acheives the highest scores among all inference methods.

Our method consistently achieves the highest average scores across both base models, demonstrating superior visual fidelity. These results confirm that, beyond improving textual metrics, our approach delivers the strongest real-world performance for Img2LaTeX by effectively handling stylistic variations in LaTeX code.

## 10   CONCLUSION

In this work, we introduce $A^2R^2$, a framework that integrates visual reasoning with attention localization and iterative refinement to enhance the performance of vision-language models (VLMs) on the Img2LaTeX task. However, several limitations remain: (1) Due to the closed-source nature of models such as GPT-4O and CLAUDE-3.5-SONNET (OpenAI et al., 2024c; Anthropic, 2024), we are unable to access their internal attention weights, which limits our ability to fully assess the effectiveness of the proposed method on these platforms. (2) Computational constraints prevent us from incorporating larger models such as LLAMA-3.2-90B-VISION-INSTRUCT in our experiments. We leave the exploration of scaling our approach to such models for future work. (3) Our current approach depends on manually crafted prompts to effectively guide model behavior. This suggests that future research could focus on developing prompt-free or prompt-agnostic methods to improve generalizability and ease of deployment.

ETHICS STATEMENT

Ethical considerations are of utmost importance in our research. In this paper, we strictly adhere to ethical principles by exclusively utilizing open-source datasets and employing models that are either open-source or widely recognized within the scientific community. Our findings highlight the strong potential for improving vision–language models on the Img2LaTeX task. We remain committed to upholding ethical standards throughout the research process, prioritizing transparency, and promoting the responsible use of technology for the betterment of society.

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

## A    THE USE OF LARGE LANGUAGE MODELS (LLMS)

We used LLMs to assist with the phrasing and grammar of the manuscript. The LLMs were used strictly as a writing aid and did not contribute to the scientific ideation, methodology, or results presented in this paper.

## B    FURTHER RELATED WORK

Here, we further discuss related work on the potential of vision–language models (VLMs) in the Img2LaTeX task.

### B.1    VISION LANGUAGE MODELS

Vision–language models (VLMs) aim to unify visual perception with linguistic understanding. Their development has been primarily driven by the emergence of Transformer architectures and contrastive learning techniques, which align visual and textual representations within a shared semantic space (Vaswani et al., 2023; Chen et al., 2020; Radford et al., 2021). VLMs are explicitly designed to process and integrate multimodal inputs, enabling strong performance across a broad range of tasks, including image captioning, visual question answering, and related vision-language understanding tasks (Liu et al., 2023; OpenAI et al., 2024b; Caffagni et al., 2024; Zhang et al., 2024a; Yin et al., 2024; Li et al., 2023; Bordes et al., 2024; Li et al., 2022; Alayrac et al., 2022). Recent advances in scaling strategies, including increases in model capacity and data volume, combined with the availability of large-scale multimodal datasets, substantially improve generalization and task-specific accuracy (Bai et al., 2025; Team et al., 2024; Li et al., 2024; Schuhmann et al., 2022; Team et al., 2025; Chen et al., 2024b; 2025; Zhang et al., 2024c). State-of-the-art models now demonstrate robust performance across diverse benchmarks, highlighting the rapid progress and increasing potential of VLMs in addressing complex multimodal tasks. However, the potential of VLMs in the Img2LaTeX task remains underexplored. While prior work has evaluated the capabilities of recent high-performing models in this domain, experimental results indicate that their performance falls short of human expectations.

## C    IMG2LATEX-HARD-1K: CONSTRUCTION PIPELINE

We present the full pipeline for constructing the *Img2LaTex-Hard-1K* dataset, which consists of three stages described below:

### C.1    MODEL GENERATION

Our filtering methodology requires comprehensive evaluation across diverse architectural paradigms and model scales. To this end, we select three representative open-weight VLMs that span the current performance spectrum:

- QWEN2.5-VL-7B-INSTRUCT and QWEN2.5-VL-32B-INSTRUCT (Bai et al., 2025): Representing the QWEN2.5 family, these models enable controlled scale comparisons within a consistent architecture.
- LLAMA-3.2-11B-VISION-INSTRUCT (Meta AI, 2024): A member of the LLAMA3.2 family, this model introduces architectural diversity and serves as a cross-family reference point.

This selection strategy ensures that our difficulty assessment reflects performance variations due to both architectural diversity and model scale, mitigating biases toward any single design paradigm.

We conduct inference with each model on approximately 7,000 instances from our dataset. For every prediction, we compute three evaluation metrics: m-ROUGE, BLEU-4, and Edit Distance. As a result, each instance is associated with nine evaluation scores, corresponding to the three metrics computed across the three models, which facilitates a comprehensive analysis of model behavior.

Let each instance in the dataset be denoted as $x_i$, where $i = 1, 2, \ldots, N$. We evaluate each instance using three models, $M_1, M_2, M_3$, corresponding to the previously described systems. For each model $M_j$ ($j = 1, 2, 3$), we compute the following evaluation metrics:

- $R_{ij}$: the m-ROUGE score for instance $x_i$ under model $M_j$.
- $B_{ij}$: the BLEU-4 score for instance $x_i$ under model $M_j$.
- $D_{ij}$: the raw Edit Distance for instance $x_i$ under model $M_j$.

To ensure comparability across metrics, we apply min-max normalization to the Edit Distance values:

$$\tilde{D}_{ij} = \frac{D_{ij} - \min(D)}{\max(D) - \min(D)},$$

where $\min(D)$ and $\max(D)$ denote the minimum and maximum Edit Distance values computed over all instances and all models.

We define a composite score $S_{ij}$ for each instance-model pair as a weighted average of the three metrics, where the normalized inverse Edit Distance $(1 - \tilde{D}_{ij})$ is treated as a positive indicator of similarity:

$$S_{ij} = \alpha \cdot R_{ij} + \beta \cdot B_{ij} + \gamma \cdot (1 - \tilde{D}_{ij})$$

We set the weights to $\alpha = 0.4$, $\beta = 0.4$, and $\gamma = 0.2$. This formulation yields a unified measure of model performance that balances lexical similarity and character-level accuracy.

## C.2 GPT EVALUATION

While textual metrics capture semantic similarity, they may fail to reflect visual rendering similarities that impact practical usability. For example, two LaTeX expressions may differ semantically yet produce visually similar outputs due to variations in generation patterns. To account for this visual dimension, we employ GPT-4O-MINI (OpenAI, 2024) as an image-level comparator, selected for its strong visual reasoning capabilities and cost-effectiveness in large-scale evaluation.

For each instance $x_i$, we generate three LaTeX code predictions $L_{ij}$ from the models $M_j$ ($j = 1, 2, 3$). Each LaTeX string is compiled into a rendered image $I_{ij}^{\text{gen}}$ using tools such as *pdflatex* and *ImageMagick*. These generated images are then compared to the ground-truth rendering $I_i^{\text{gt}}$ to assess visual fidelity.

To evaluate reproduction accuracy, we employ the GPT-4O-MINI model as an image comparator. For each pair $(I_i^{\text{gt}}, I_{ij}^{\text{gen}})$, we prompt GPT-4O-MINI to assign a similarity score $G_{ij} \in [0, 10]$, indicating how faithfully the generated LaTeX output reproduces the visual content of the original image.

To align this visual fidelity score with the other evaluation metrics, which lie in the range $[0, 1]$, we apply linear normalization:

$$\tilde{G}_{ij} = \frac{G_{ij}}{10},$$

where $\tilde{G}_{ij} \in [0, 1]$ denotes the normalized visual reproduction score for instance $x_i$ under model $M_j$. This normalization enables fair integration of visual fidelity into the unified evaluation framework.

## C.3 DATA SELECTION

After computing four evaluation scores for each model's output on each input instance, we assign weights to the models based on their parameter scales and empirical performance. Specifically, we set the model weights as follows:

$$w_1 = 0.30 \quad (\text{QWEN2.5-VL-7B-INSTRUCT})$$
$$w_2 = 0.40 \quad (\text{QWEN2.5-VL-32B-INSTRUCT})$$
$$w_3 = 0.30 \quad (\text{LLAMA-3.2-11B-VISION-INSTRUCT})$$

Following the metric and visual fidelity computations described above, we define the final score for instance $x_i$ under model $M_j$ as:

$$S_{ij}^{\text{final}} = S_{ij} + 0.5 \cdot \tilde{G}_{ij},$$

where $S_{ij}$ is the composite textual metric score and $\tilde{G}_{ij}$ is the normalized GPT-4O-MINI visual fidelity score. To aggregate model-specific evaluations into a single score per instance, we compute a weighted sum:

$$S_i^{\text{final}} = \sum_{j=1}^{3} w_j \cdot S_{ij}^{\text{final}}$$

We then rank all instances by $S_i^{\text{final}}$ in descending order and select the top 1,100 examples to construct the *Img2LaTex-Hard-1K* benchmark. This subset is intended to better reflect the evaluation demands of modern, high-capacity models.

## D  IMPLEMENTATION DETAILS

We follow established methodologies for attention-based localization as proposed in prior work (Yao et al., 2025; Li et al., 2025b). For LLAMA-3.2-11B-VISION-INSTRUCT, we extract attention maps from the 13th cross-attention layer, as cross-attention is applied every five layers between the 3rd and 38th layers. This layer has been shown to effectively capture cross-modal interactions, particularly in OCR-focused tasks. For QWEN2.5-VL-32B-INSTRUCT, we compute the mean attention across the central one-eighth of all layers, which serves as a representative proxy for identifying salient regions relevant to our localization objective.

## E  BASELINE INTRODUCTION

In the main experiments, we compare our proposed method against three baseline approaches, which are briefly described below:

- *Direct Prompting:* Generates LaTeX code directly from the input prompt and image without any auxiliary reasoning or guidance.
- *Chain-of-Thought (CoT) Prompting:* Implements zero-shot CoT prompting by appending the phrase *"Let's think step by step"* to the input prompt, encouraging the model to decompose the problem into intermediate reasoning steps (Kojima et al., 2023; Wei et al., 2023).
- *Best-of-N:* Generates $N$ candidate LaTeX sequences in parallel and selects the one that best satisfies a predefined verification metric.

