# OpenReview forum: "$A^2R^2$: Advancing Img2LaTeX Conversion via Visual Reasoning with Attention-Guided Refinement"
_ICLR.cc/2026/Conference — ICLR 2026 Conference Withdrawn Submission_

### Official Review · Reviewer_D25D · 2025-11-01

**Soundness:** 3
**Presentation:** 3
**Contribution:** 2
**Rating:** 4
**Confidence:** 4

**Summary:**

This paper introduces **A2R2**, a framework designed to improve **Img2LaTeX conversion** by integrating **visual reasoning** and **attention-guided refinement**.
The Img2LaTeX task—converting images of mathematical expressions into LaTeX code—remains highly challenging, especially when handling **fine-grained visual components** such as subscripts, superscripts, and nested structures.
To address these challenges, A2R2 adopts a **multi-stage iterative reasoning process**, which refines LaTeX predictions through **attention localization** and **feedback verification**, allowing the model to iteratively correct its own outputs.

In addition, the paper introduces a new benchmark, **Img2LaTeX-Hard-1K**, which contains **1,100 carefully curated and challenging examples** that test the limits of LaTeX generation on **complex visual inputs**.

Empirically, **A2R2 significantly outperforms existing models and baselines** across multiple **textual and visual evaluation metrics**, including **ROUGE**, **BLEU**, and **CW-SSIM**.
The framework also shows **consistent performance improvements as the number of inference rounds increases**, demonstrating its effectiveness in **test-time scaling** and iterative visual reasoning.

**Strengths:**

1. The paper focuses on the relatively niche yet technically challenging **Img2LaTeX** problem.
   This clear and well-scoped objective allows for a deep exploration of visual–symbolic reasoning within a specific and meaningful application domain.

2.
   The release of **Img2LaTeX-Hard-1K**, a dataset specifically curated for **hard and visually complex mathematical expressions**, provides a **valuable new benchmark** for future research in **formula recognition** and **symbolic reasoning**.
   This resource enhances both the reproducibility and long-term impact of the work.

3.
   By first **rendering the predicted LaTeX** and then feeding it back to the VLM for verification, the paper embodies the insight that **generation can enhance understanding**.
   This approach offers an elegant and empirically supported example of **self-reflective reasoning** in multimodal generation systems.

**Weaknesses:**

1. **Potentially Misleading Terminology (“Attention-Guided Refinement”)**
   The term *“Attention-Guided Refinement”* is somewhat misleading, as the proposed framework does **not include any actual attention-layer modeling**.
   Instead, it relies purely on **semantic-level comparison** between the generated LaTeX and the rendered image.
   A more precise term would better reflect the underlying mechanism and avoid confusion with attention-based architectures.

2. **Lack of Novel Data Contribution**
   Despite positioning itself as a **data-centric and training-free** approach, the work does **not introduce any new dataset**.
   The proposed *Img2LaTeX-Hard-1K* benchmark is simply a **sampled subset (1K cases)** from an existing and rather dated dataset (circa 2017).
   This weakens the originality and long-term research value of the data contribution.

3. **Missing Comparative Training Baselines (DPO/GRPO)**
   Given the presence of an explicit **verifier mechanism**, it would be natural to include **DPO or GRPO**-based experiments as comparisons.  Without such baselines, the paper’s contribution appears limited to a **prompt-based iterative engineering trick** that slightly improves accuracy on a niche task, rather than a fully developed research framework suitable for formal publication.

**Questions:**

Given that the paper already incorporates a clear verifier-based design, why didn’t the authors include DPO or GRPO experiments as comparative baselines?

---

### Official Review · Reviewer_xsXM · 2025-11-01

**Soundness:** 2
**Presentation:** 2
**Contribution:** 2
**Rating:** 4
**Confidence:** 5

**Summary:**

This paper proposes A2R2, a training-free, plug-and-play method designed to enhance MLLMs’ image-to-LaTeX (img2latex) capabilities. The method relies on a render–compare–refine loop. The authors also introduce a new benchmark of 1K samples to better evaluate img2latex performance. Experimental results demonstrate improvements when applying A2R2 to multiple MLLMs.

**Strengths:**

Clear writing and presentation. The paper is well-organized, with high-quality figures and clear explanations that make the method easy to follow.

Demonstrated effectiveness. Results on the proposed benchmark show that the approach consistently improves performance.

Comprehensive analysis. The experimental section includes detailed ablations and analysis.

**Weaknesses:**

Limited novelty. The render–compare–refine pipeline is intuitive and analogous to well-known strategies in text-to-image and code-generation settings. While practical, it offers limited conceptual novelty and lacks deeper algorithmic insight. Without a training component, the method reads as an incremental engineering improvement.

High inference-time overhead. Although training-free, the method introduces multiple iterative steps, which could lead to substantial test-time cost. A comparison of inference-time efficiency versus baselines (e.g., best-of-N sampling) is necessary to justify the trade-off between performance gains and computational overhead.

Benchmark motivation unclear. The new benchmark appears to primarily sample from existing datasets. The motivation for a separate benchmark and its unique value beyond existing ones are not clearly articulated. More justification—or inclusion of additional challenging elements—would strengthen this contribution.

**Questions:**

none

---

### Official Review · Reviewer_msHm · 2025-11-02

**Soundness:** 3
**Presentation:** 3
**Contribution:** 3
**Rating:** 4
**Confidence:** 5

**Summary:**

This paper addresses the task of Img2LaTeX, which involves converting images of mathematical expressions into their corresponding LaTeX source code. The authors observe that while modern VLMs show promise, they often fail on fine-grained details like subscripts, superscripts, and complex structures. To tackle this, they propose A²R², a novel, training-free inference framework that enables VLMs to perform iterative self-correction. The core idea is a multi-step reasoning loop: (1) Generate an initial LaTeX hypothesis, (2) Render it into an image, (3) Compare the rendered image with the original input to identify discrepancies, (4) Use attention maps to localize these mismatched regions and verify the feedback, and (5) Refine the LaTeX code based on the verified, localized feedback. This process can be repeated to progressively improve the output. To facilitate a more rigorous evaluation, the authors also introduce Img2LaTeX-Hard-1K, a new challenging benchmark of 1,100 samples curated from the Im2LaTeX-100K dataset. Extensive experiments on two open-weight VLMs (LLaMA-3.2-11B and Qwen2.5-VL-32B) demonstrate that A²R² significantly outperforms baseline methods like direct prompting, Chain-of-Thought, and Best-of-N sampling across a suite of textual and visual metrics.

**Strengths:**

1. The proposed A²R² framework is well-designed and conceptually elegant. The idea of "closing the loop" by having the model render its own output and visually compare it against the input is a powerful form of self-verification.

2.  The paper presents compelling evidence for the effectiveness of A²R². The framework consistently and significantly outperforms all baselines across two different model architectures and scales.

3. The introduction of the Img2LaTeX-Hard-1K dataset is a significant contribution to the community.

**Weaknesses:**

1. The primary drawback of the A²R² framework is its significant computational cost. Each refinement cycle involves multiple VLM inference calls (Comparison, Verification, Refinement) plus the overhead of an external rendering tool. While the authors cap the rounds at two for a fair comparison with Best-of-N=8 in terms of token count, the sequential nature of the A²R² loop will inevitably lead to much higher wall-clock latency.

2. The paper frames A²R² as a purely inference-time, training-free method. While effective, this approach acts as an external "scaffold" to guide a flawed base model, rather than fundamentally improving the model's intrinsic capabilities. A potentially more impactful and efficient application of this framework would be to leverage its components during training.

**Questions:**

Could you provide a more direct analysis of the inference overhead?

---

### Note · Authors · 2025-12-01

I have read and agree with the venue's withdrawal policy on behalf of myself and my co-authors.